# A Simple 1D Convection-Diffusion Model of Oxalic Acid Oxidation Using Reactive Electrochemical Membrane

**DOI:** 10.3390/membranes11060431

**Published:** 2021-06-07

**Authors:** Ekaterina Skolotneva, Marc Cretin, Semyon Mareev

**Affiliations:** 1Physical Chemistry Department, Kuban State University, 149 Stavropolskaya str, 350040 Krasnodar, Russia; ek.skolotneva@gmail.com; 2Institut Europeen des Membranes, IEM-UMR 5635, ENSCM, CNRS, Université Montpellier, 34095 Montpellier, France; marc.cretin@umontpellier.fr

**Keywords:** reactive electrochemical membrane, porous electrode, anodic oxidation, hydroxyl radicals

## Abstract

In recent years, electrochemical methods utilizing reactive electrochemical membranes (REM) have been recognized as the most promising technologies for the removal of organic pollutants from water. In this paper, we propose a 1D convection-diffusion-reaction model concerning the transport and oxidation of oxalic acid (*OA*) and oxygen evolution in the flow-through electrochemical oxidation system with REM. It allows the determination of unknown parameters of the system by treatment of experimental data and predicts the behavior of the electrolysis setup. There is a good agreement in calculated and experimental data at different transmembrane pressures and initial concentrations of *OA*. The model provides an understanding of the processes occurring in the system and gives the concentration, current density, potential, and overpotential distributions in REM. The dispersion coefficient was determined as a fitting parameter and it is in good agreement with literary data for similar REMs. It is shown that the oxygen evolution reaction plays an important role in the process even under the kinetic limit, and its contribution decreases with increasing total organic carbon flux through the REM.

## 1. Introduction

According to the UN WWDR [1], water quality management is one of the main environmental problems of humankind. There are a lot of biorefractory toxic organic pollutants, the removal of which requires the implementation of novel wastewater treatment and drinking water production systems. Anodic oxidation (AO) is an electrochemical advanced oxidation process (EAOP) that is increasingly recognized as a promising next-generation technology for the treatment of contaminated effluents. [2,3,4,5]. This process is based on the removal of organic pollutants by a combination of direct electron transfer from the contaminate (*R*) to the electrode (Equation (1)) and the generation of a large amount of highly reactive hydroxyl radicals (HO^•^) from the water discharge on the surface of the electrode (*S*), which has a high oxygen overpotential (Equation (2)) [6,7].
(1)$$R\to R^{•}+\bar{e}$$
(2)$$S+H_{2}O\to S\left({\mathrm{HO}}^{•}\right)+H^{+}+\bar{e}$$

Recent studies have demonstrated that the mineralization of a large number of biorefractory organic pollutants can be achieved using AO [8,9,10,11,12,13,14]. However, several scientific challenges remain to be overcome to promote the widespread application of AO for water treatment. The main problem is the diffusion limitations, which lead to low mass transport of pollutants comparing to the oxidation rate [15]. Hydroxyl radicals are formed only on the anode surface and have a short lifetime; therefore, they are present only in a thin boundary layer (<<1 μm) [16,17]. As a result, the oxidation of organic pollutants occurs on the electrode surface, and when the current density reaches the kinetic limiting value, the process is restricted by convective-diffusion delivery of pollutants from the solution to the reaction zone. The conventional anodes are installed in parallel plate reactors, in which mass transfer is strongly limited by diffusion through a thick (~0.1–1 mm) stagnant boundary layer. Recent studies have shown that the most effective way to avoid diffusion limitations is to use the reactive electrochemical membranes in flow-through configuration, that is, the solution is pumped through the anode [18,19,20].

A quantitative description and determination of the optimal parameters of the oxidation process are possible only when an adequate mathematical model is applied. Currently, in the literature, there are a large number of different models of anodic oxidation processes on conventional plate electrodes. In the study of Comninellis [6], two cases of oxidation are considered: on active and passive anodes, for each of which a relation for instantaneous current efficiency is derived. Simond and coauthors [21] have extended this model: they took into account adsorption in the equations for heterogeneous reaction rates and defined the effectiveness factor to quantify the decrease in the current efficiency due to concentration polarization. Panizza et al. [22] presented a very simple macroscopic model describing chemical oxygen demand (COD) dependence on time, which allows estimating the electrical charge required to remove the given COD value. Scialdone [23] has developed this approach by adding the so-called “mixed regime” and taking into account the concentrations of different species instead of the global COD parameter. Further, Lan and coauthors [24] improved this model by taking into account the direct electron transfer. Canizares et al. [25] developed a model in which concentration profiles for all organic compounds were derived by assuming that the electrochemical reactor can be represented as three interconnected zones and in each of its zone, for simplicity, the concentration of each compound is assumed to be time-dependent and position-independent. Mascia and coauthors [26] presented a 1D model predicting the impact of operating conditions on the efficiency of the anodic oxidation process. In the study of Kapalka et al. [16], the analytical expression for the spatial distribution of hydroxyl radicals concentration in the presence and the absence of organic compounds, which allows calculating the reaction zone thickness, was derived. Groenen-Serrano and coauthors [27] studied the competitive oxidation of two organic compounds on the BDD anode surface using the 1D non-stationary model.

Currently, works on the anodic oxidation of organic pollutants using REM are aimed primarily at the experimental study. The mathematical modeling in this area is very poorly represented. Jing and coauthors developed a mathematical transmission line model for estimation of the active layer thickness and REM fouling [28]. Misal and coauthors used the one-dimensional model to estimate the potential and current density distribution and the evolution of the concentration of organic compounds within REM [29]. In our recent works [30,31], we used models concerning the transport and reaction of organic species with hydroxyl radicals generated at a TiOx REM operated in a flow-through mode to obtain the dependences of the paracetamol mineralization efficiency on the pore radius and porosity of REM [30] and the oxygen bubbles formation [31].

In this study, we present the theoretical analysis based on a one-dimensional model of a flow-through anodic oxidation system. This model takes into account both the organic compound oxidation and oxygen evolution reaction. It was calibrated based on experimental data obtained for the oxidation process of oxalic acid in the system with sub-stoichiometric titanium oxide REM.

## 2. Mathematical Model

### 2.1. The Geometry of the System under Study

We consider a cross-flow electrolyzer, which utilizes REM as a porous anode in inside-outside cross-flow filtration mode. The experimental setup is described in detail in [5]. The REM is a porous electrode (in our case, it is a tubular sub-stoichiometric titanium oxide electrode with a wall thickness of 2 mm). The transmembrane pressure (TMP) was used as an independent parameter of the experiment. The feed solution contains an oxalic acid (*OA*) of varying concentrations, which is the target pollutant, and a supporting electrolyte (0.1 M Na_2_SO_4_), which presumably does not participate in any chemical reactions and is used only to decrease the total resistance of the system. All the calculations were performed in galvanostatic mode.

The system under study consists of REM (anode) with the adjacent diffusion layer (DL) of thickness *δ* (Figure 1). The solution of initial concentration *c*_0_ flows from the bulk solution (inlet), through the pores of REM to the permeate. The cathode is placed in the bulk solution, thus the current in the considered system passes from the right (*x* = *d*) to the left (*x* = −*δ*).

### 2.2. The Problem Formulation

According to the previous theoretical investigations [16,32,33], the following simplifying assumptions are made: The transport number of organic compound is negligible compared to the transport number of the supporting electrolyte. Thus, only diffusion and convection fluxes are considered;The system under study is in a steady state, thus only the faradaic current is taken into account;Since the experiment proceeds under room conditions and the supporting electrolyte does not participate in the reactions, the gradients of temperature, activity coefficients, and density are ignored;The oxygen concentration in the solution exceeds the solubility limit only at the lowest *TOC fluxes* and is insufficient at given current density [31]; thus, the bubble-formation caused by oxygen evolution is not taken into account;The rate constant of oxalic acid oxidation by hydroxyl radicals is very small, thus, we assume that all the hydroxyl radicals are spent on the oxygen molecules formation;According to the conditions of the experiment [5], the bulk solution is considered perfectly mixed and renewable, so the oxalic acid concentration in it is assumed constant.

In our previous work [30], a 2D model of the transport of diluted species in a system similar to that presented in this work was used to describe the process of anodic oxidation of organic pollutants. The pore shape was considered cylindrical throughout the REM depth, the direct electron transfer (*DET*) of organic species and oxygen evolution reactions were not taken into account, and the electrode conductivity was considered to be significantly higher than that of the solution. These assumptions are valid when considering the processes occurring at the interface between the diffusion layer and the electrode. In [30], the processes occurring at a depth of REM from 0 to 30 µm from its surface were considered, and the concentration of organic compound (paracetamol) in all calculations was extremely low (0.19 mM), which makes it possible to use the above assumptions. In our recent work [31], it was shown that during the simulation of anodic oxidation of organics, it is necessary to take into account the oxygen evolution reaction, since in the case of Magnelli phases of stoichiometric titanium oxide even at low current densities, the contribution of the reaction to electron transfer is very high. When considering the entire volume of REM, 2D simulation is difficult due to the need for large computing power, which extremely complicates the calculations. To simplify the system under study, we use a 1D model.

Based on previous studies related to the porous electrodes [34,35], a reactive transport model was developed to investigate the electrochemical oxidation of *OA* in the REM reactor. The transport of diluted species in the solution is described by the equation system, which consists of Fick’s law with the convective term (3), material balance Equation (4), Ohm’s law in the differential form written for each of the phases (5)–(6), charge conservation law (7)–(8), and Darcy’s law (9):(3)$$J_{k}=-\varepsilon_{s}\left(D_{k}+D_{k}^{a}\right)\frac{\partial c_{k}}{\partial x}+c_{k}v$$
(4)$$\varepsilon_{s}a_{v}J_{k}^{n}-\frac{dJ_{k}}{dx}=0$$
(5)$$i_{s}=-\varepsilon_{s}^{}\kappa_{s}\frac{\partial \varphi_{s}}{\partial x}$$
(6)$$i_{m}=-\varepsilon_{m}\kappa_{m}\frac{\partial \varphi_{m}}{\partial x}$$
(7)$$\frac{\partial \left(i_{m}\right)}{\partial x}=\varepsilon_{m}\overset{2}{\underset{k=1}{\sum }}a_{v}i_{k}$$
(8)$$\frac{\partial \left(i_{s}\right)}{\partial x}=-\varepsilon_{s}\overset{2}{\underset{k=1}{\sum }}a_{v}i_{k}$$
(9)$$v=\frac{\sigma }{\mu }\frac{\mathrm{TMP}}{d}$$
where ${\vec{J}}_{k}$, $D_{k}$, and *c_k_* are the flux density, diffusion coefficient, and concentration of the *k*th component of the system, respectively; $i_{j}$, $\kappa_{j}$, $\varepsilon_{j}$ and $\varphi_{j}$ are the current density, conductivity, volume fraction and potential of the solution (*j* = *s*) and the anode material (*j* = *m*); v is the total velocity of permeate; *σ* is the permeability coefficient; *μ* is the dynamic viscosity; *i_k_* is the local current density of oxidation of species *k*; *a_v_* is the specific surface area of the electrode; $J_{k}^{n}=k_{m}\left(c_{k}^{w}-c_{k}^{}\right)$ is the local pore-wall flux of the *k*th species to the flowing solution, which is related to the average mass-transfer coefficient, *k_m_* [35]; $c_{k}^{w}$ is the concentration on the wall. *φ*, *i*, *J* and *c* are functions of *x*. Equations (3) and (4) describe the concentration field, and Equations (5) and (8) describe the potential field and electric current distribution.

In [36], the authors state that titanium oxide should have a propensity to form HO radicals as a product of the water discharge on the surface of an anode. According to Weiss and coauthors [37], the absolute rate constant for the reaction of *OA* oxidation by hydroxyl radicals is very low—1.4 × 10^−3^ mol/(m^3^·s), thus, in our case, faster mineralization of AO can be achieved by *DET*. The generated HO^•^ radicals are consumed in the recombination reaction and form the oxygen molecule. Thus, the total oxygen evolution reaction proceeds preferably during the *DET* and also during the recombination of HO^•^. We consider the following oxygen evolution reaction:(10)$$2H_{2}O\overset{\;DET\;}{\to }O_{2}+4H^{+}+4e$$

The oxidation of *OA* by *DET* proceeds in parallel to the reaction (10):(11)$$OA+2H_{2}O\overset{\;DET\;}{\to }2{\mathrm{CO}}_{2}+2H_{2}O+H^{+}+2\bar{e}$$

The current density spent on the *DET* reactions (10) and (11) can be expressed using the Butler-Volmer Equation:(12)$$i_{OA}=i_{0}^{OA}\frac{c_{OA}^{w}}{c_{0}^{OA}}\exp\left[\frac{\beta_{OA}n_{OA}F}{RT}\eta_{}^{OA}\right]$$
(13)$$i_{O_{2}}=i_{0}^{O_{2}}\exp\left[\frac{\beta_{O_{2}}n_{O_{2}}F}{RT}\eta_{}^{O_{2}}\right]$$
(14)$$\eta_{}^{k}=\varphi_{m}-\varphi_{s}-E_{k}^{0}$$
where $i_{0}^{k}$, $n_{k}$, $\eta_{}^{k}$, $E_{k}^{0}$, $\beta_{k}$ are the exchange current density, number of electrons, overpotential, formal electrode potential, and the electron transferred coefficient of the *k*th species, where *k* takes the values of *OA* and O_2_ in reactions (10) and (11), respectively; *F* is the Faraday constant; *R* is the gas constant; *T* is the temperature. 

With the assumption that the *OA* participates only in the reaction (11), we have:(15)$$i_{OA}=-\frac{n_{OA}F}{s_{OA}}J_{OA}^{n}$$

The substitution of Equation (15) into (12) yields:(16)$$-\frac{n_{OA}F}{s_{OA}}J_{OA}^{n}=i_{0}^{OA}\frac{c_{OA}^{w}}{c_{0}^{OA}}\exp\left[\frac{\beta n_{OA}F}{RT}\eta_{}^{OA}\right]$$

Solving the Equation (16) for the $J_{OA}^{n}$ taking into account that $c_{OA}^{w}=\frac{J_{OA}^{n}}{k_{m}}+c_{OA}^{}$ with subsequent substitution into (4) and into (15) gives (17) and (18), respectively:(17)$$\varepsilon_{s}\left(D_{OA}+D_{OA}^{a}\right)\frac{d^{2}c_{OA}}{dx^{2}}-v\frac{dc_{OA}}{dx}=\varepsilon_{s}a_{v}\frac{c_{OA}}{\frac{1}{k_{m}}+\frac{n_{OA}Fc_{0}^{OA}}{s_{OA}i_{0}^{OA}}\exp\left[\frac{-\beta_{OA}n_{OA}F}{RT}\eta_{}^{OA}\right]}$$
(18)$$i_{OA}^{}=\frac{c_{OA}}{\frac{s_{OA}}{n_{OA}Fk_{m}}+\frac{c_{0}^{OA}}{i_{0}^{OA}}\exp\left[\frac{-\beta_{OA}n_{OA}F}{RT}\eta_{}^{OA}\right]}$$

With the assumption that the pore-wall flux of O_2_ to the flowing solution depends only on the overpotential value and does not affect the rates of the chemical reactions, we can write:(19)$$J_{O_{2}}^{n}=-\frac{s_{O_{2}}i_{O_{2}}}{n_{OA}F}$$

The substitution of (3), (13), and (19) into (4) gives:(20)$$\varepsilon_{s}\left(D_{O_{2}}+D_{O_{2}}^{a}\right)\frac{d^{2}c_{O_{2}}}{dx^{2}}-v\frac{dc_{O_{2}}}{dx}=\varepsilon_{s}a_{v}\frac{s_{O_{2}}i_{0}^{O_{2}}}{n_{O_{2}}F}\exp\left[\frac{\beta_{O_{2}}n_{O_{2}}F}{RT}\eta_{}^{O_{2}}\right]$$

In the diffusion layer, Equations (17) and (20) are as follows:(21)$$D_{OA}\frac{d^{2}c_{OA}}{dx^{2}}-v\frac{dc_{OA}}{dx}=0$$
(22)$$D_{O_{2}}\frac{d^{2}c_{O_{2}}}{dx^{2}}-v\frac{dc_{O_{2}}}{dx}=0$$

According to Equations (7) and (8), the sum of current densities (*i_m_* + *i_s_*) in the system is assumed constant and equal to *i_tot_*.

In the bulk of the solution (x=−δ), the concentration of *OA* and electrolyte potential are set constant:(23)$$c_{OA}=c_{0}$$
(24)$$\varphi_{s}=0$$

In the inlet of the REM (x=0), the current density in the electrode material phase is equal to zero:(25)$$i_{m}=0$$

In the outlet of the REM (x=−d), the permeate flux is equal to the convective term and the current density in the electrode material phase is equal to the total one. Thus, we can write the following boundary conditions:(26)$$\frac{dc_{OA}}{dx}=0$$
(27)$$\frac{dc_{O_{2}}}{dx}=0$$
(28)$$i_{m}=i_{tot}$$

## 3. Results and Discussion

The problem is solved numerically using Comsol Multiphysics 5.5 software package.

### 3.1. The Treatment of Experimental Data

According to previous studies [38,39], hydroxyl radicals are able to participate in the oxidation of *OA*. However, the absolute rate constant for the reaction of HO^•^ with *OA* is very low, 1.4 × 10^3^ m^3^/mol s, and Guo et al. [40] considered that *OA* could be used as a *DET* oxidation probe. Thus, in this paper, the reaction of *OA* oxidation by HO^•^ is not considered.

The experiment was carried out in our previous work [5]. The parameters of the experimental setup and model ones are presented in Table 1. The REM was a Magnelly phase TiOx anode with monomodal pore size. 

The REM parameters have been investigated in detail in [5]. It should be noted that the permeability coefficient was determined without taking into account the electric current. After permeate flux measurements, it was found that when an electric current is applied, the solution flux increases due to the electroosmotic flow. The calculation results showed that its contribution is about 12 percent at TMP = 40 mbar [31]. In the current work, the value of the parameter *σ* is assumed constant and equal to 1.7 × 10^−14^ m^2^.

The value of *δ* = 30 μm is obtained using the Lévêque approximate solution for the hydrodynamic conditions and geometric parameters of the electrolyzer used in the experiments [30].

The fitting of the calculated and experimental data was carried out through optimization of dispersion coefficient, *D^a^*, and formal potential in reaction (11), $E_{OA}^{0}$. *D^a^* primarily affects the value of the OA flux in the solution of electrode pores and $E_{OA}^{0}$ affects its oxidation during *DET*. The dispersion coefficient makes the concentration distribution in the electrode pore solution more uniform, which primarily affects the calculation data at high concentrations and low flow rates. The value of the formal potential affects the oxygen evolution reaction rate. In our case, it affects the rate of *OA* oxidation, because the total current is constant. The obtained value of *D_a_* is in good agreement with the data presented in [29] for the case of flat Magnelly phase TiOx REM.

The calculated data was fitted to the experimental ones using the parameters in Table 1 in both cases: at a variable concentration and constant TMP = 40 mbar, and a variable TMP and constant concentration, *c*_0_ = 18 mgC/L (Figure 2).

The following performance indicators were used for the determination of the oxidation efficiency: the total organic compound (*TOC*) flux, percentage of removal (*PR*), and mineralization current efficiency (*MCE*, percentage of current directed towards the mineralization of the substrate passing through the REM):(29)$$TOC\;flux=c_{0}nJ_{OA}$$
(30)$$PR=\frac{\left(c_{0}-c_{outlet}\right)}{c_{0}}\times 100\%$$
(31)$$MCE=\frac{n_{OA}FJ_{OA}\left(c_{0}-c_{outlet}\right)}{3600i_{tot}}\times 100\%$$
where *n* is the mass of organic carbon in one mol of *OA* (24 g/mol); $J_{OA}$ is the permeate flow of *OA* (m^3^/(h × m^2^)); $c_{outlet}$ is the permeate concentration (mol/m^3^).

### 3.2. Effect of Increasing Concentration at a Constant Transmembrane Pressure 

Figure 2 presents the calculated and experimental data obtained for the two-electron oxidation of *OA* at −150 A/m^2^. From the data obtained, it follows that the percentage removal and mineralization efficiency are influenced by both the *OA* concentration and the permeate flow. When the experiments were carried out at a constant TMP value (40 mbar) and increasing *OA* concentrations (*c*_0_ = 18–800 mgC/L), the *MCE* value tends to plateau and reaches about 72%, respectively, which indicates that the system is approaching kinetic restriction. At the lowest *OA* flux (<30 g/m^2^ h, TMP = 40 mbar and *c*_0_ = 18 mgC/L), almost complete *OA* mineralization was achieved, and it can be seen at calculated concentration distributions (Figure 3). However, at the highest *OA* flux, the kinetic limitation of *OA* oxidation is observed, which did not allow reaching the maximum process efficiency. This is due to the fact that some of the electrons are also consumed in the oxygen evolution reaction (including HO^•^ recombination reaction).

### 3.3. Effect of Increasing Transmembrane Pressure at Constant Concentration

A different behavior was observed at constant *OA* concentration (*c*_0_ = 18 mgC/L) and increasing TMP (i.e., increasing permeate flow). In the experimental data, the *MCE* of *OA* followed a bell-shaped curve with a maximum of 21% achieved at a *TOC flux* of 29 g/m^2^ h. On the calculated data, a decrease in *MCE* with an increase in the permeate flux is not observed, therefore, the phenomena responsible for this behavior are not taken into account in the model. With an increase in the permeate flow, the oxalic acid molecules are less likely to reach the electrode surface, which in the current one-dimensional model can be expressed by a decrease in the dispersion coefficient, but in our calculations it is assumed to be constant. Oxidation of *OA* tends to the kinetic limit with an increase in the *OA* flux, similar to the case of an increase in the concentration of *OA*, but with lower total values of *MCE*. The difference between the two modes (at *c*_0_ = const and TMP = const) is the greater oxygen evolution in the case of *c*_0_ = const (Figure 4).

### 3.4. The Oxygen Evolution in Pores of the Reactive Electrochemical Membrane

Oxygen evolution decreases with an increase in *TOC flux* in both cases (at *c*_0_ = const and TMP = const) due to the growth of the *OA* molecules number in the pores of REM, which enhances the contribution of the *DET* reaction of *OA* in the resulting current density.

The oxygen evolution in the pores of REM proceeds unevenly in its bulk: it is much faster at the inlet and outlet of REM (Figure 5a,b). The *DET* reaction of *OA* is also enhanced at the REM input and output (Figure 5c,d). This behavior is observed when the electrical conductivities of the phases of a porous electrode (solution and electrode material) have similar values. In [29], during the oxidation and reduction of sulfamethoxazole at currents below the limiting kinetic value, similar dependences were observed.

In the pores of REM, the concentration of dissolved molecular oxygen increases when the oxygen evolution reaction occurs. The solubility limit of molecular oxygen at atmospheric pressure and given temperature is close to 3 mol/m^3^. Consequently, bubble formation can only be observed at the lowest *TOC fluxes* (Figure 6). With an increase in *c*_0_ and/or TMP, the oxygen concentration decreases and does not exceed the solubility limit (Figure 6). The resulting contribution of the bubbles in electrical and hydraulic resistances does not exceed the experimental measurement error.

It should be noted that at higher current densities or lower *OA* fluxes, it is mandatory to take into account the evolution of bubbles since the oxygen concentration will overcome the solubility limit. This phenomenon is considered in detail in [31].

### 3.5. The Current Density, Potential, and Overpotential Distributions

In current work, the electrical conductivities of the solution and the electrode are assumed constant, the distribution of the potential and current density in these phases depends only on the chemical reactions occurring at its interface (pore walls). The current density distribution depends very little on the *TOC flux* (<1%) and it remains almost unchanged at a constant current density (Figure 7). Since the specific conductivities of the electrode and solution are equal, and the fraction of the electrode phase is slightly larger (0.59 versus 0.41), the current density distribution in the system is asymmetric, and the point, in which *i_m_* = *i_s_*, is shifted towards the diffusion layer.

The potential distribution in the solution phase weakly depends on the *TOC flux*. The potential drop across the electrode phase also remains constant at different *TOC fluxes* (Figure 8). Only the interface (pore walls), where electrochemical reactions take place, provides a significant contribution to the resulting potential drop in the entire system. At a constant concentration value, the potential drop changes less than at a constant TMP. This is due to the concentration dependence of the *DET* reaction of *OA*, and at low concentration of organics (*c*_0_ = 18 mgC/L) the oxygen evolution reaction, which is concentration-independent in the experimental conditions, dominates in the system.

### 3.6. Effect of Current Density on Mineralization Current Efficiency 

As was expected, with the increasing total current density, the *PR* increases in both cases (at *c*_0_ = const and TMP = const). At *i_tot_* = −300 A/m^2^ even at high *OA* fluxes (70 g/m^2^ h), 99.9% removal may be achieved with 50% of *MCE* in the case of constant TMP = 40 mbar (Figure 9). At constant initial concentration *c*_0_ = 18 mgC/L, the maximum *MCE* is much lower at any *OA* flux and *i_tot_*. Nevertheless, the optimum parameters of the oxidation process may be found using the theoretical analysis at any concentration of *OA* or TMP.

## 4. Conclusions

In the paper, we proposed a 1D stationary model of transport of diluted species in the flow-through electrolysis system with the reactive electrochemical membrane. The model takes into account the geometrical, electrical, and hydrodynamic properties of the system, as well as electrochemical reactions. There are only two fitting parameters, which simplifies the calibration of the model.

It is shown that at low oxalic acid fluxes, the oxygen evolution reaction dominates in the system, but the concentration of oxygen just slightly surpasses the solubility limit. The reaction rates rise from the center of reactive electrochemical membrane bulk towards the inlet and outlet if the kinetic limit is not reached. The behavior is due to the similar values of electrode and solution phase conductivities. At the conditions close to the kinetic limit, the rate of direct electron transfer reaction of oxalic acid increases from the outlet to the inlet of the reactive electrochemical membrane.

Using a brief theoretical analysis, it was found that even at a high oxalic acid flux (70 g m^−2^ h^−1^), 99.9% percentage removal (*PR*) and 50% mineralization current efficiency (*MCE*) may be achieved at high current density (−300 A/m^2^).

## Figures and Tables

**Figure 1 membranes-11-00431-f001:**
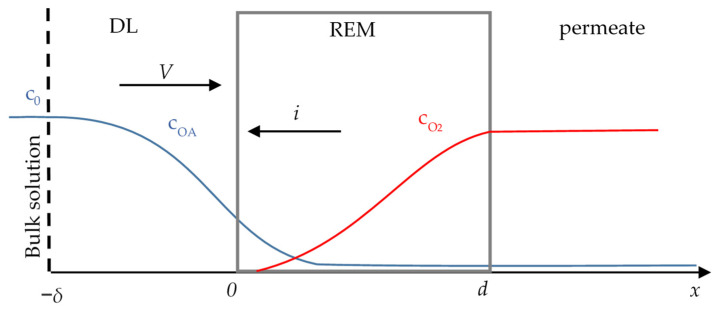
Schematic representation of the system under study.

**Figure 2 membranes-11-00431-f002:**
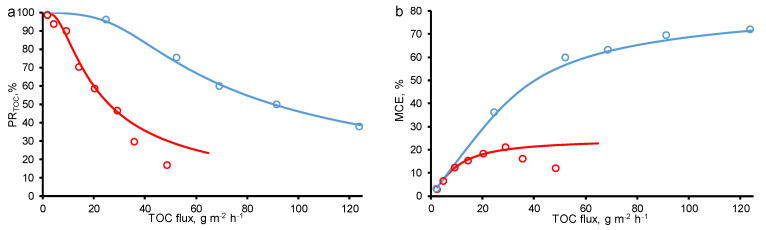
Calculated (lines) and experimental (dots) efficiency of oxalic acid removal from water by anodic oxidation on the REM as a function of total organic carbon (*TOC*) flux through the porous electrode: (**a**) percentage of TOC removal (*PR_TOC_*), (**b**) mineralization current efficiency (*MCE*). Results from increasing concentrations of oxalic acid (TMP = constant = 40 mbar, blue line) are compared with results from increasing TMP (*c*_0_ = constant = 18 mgC/L, red line). Other parameters are presented in Table 1.

**Figure 3 membranes-11-00431-f003:**
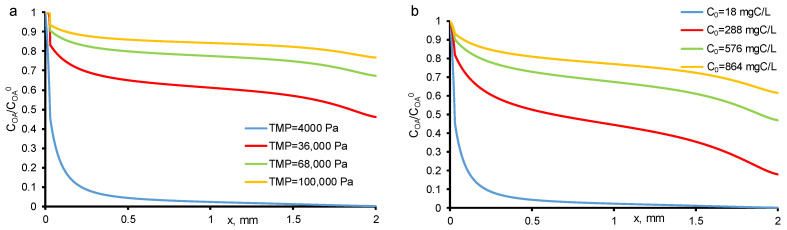
Calculated concentration distributions of oxalic acid normalized by its initial concentration (*c*_0_) in the system at different TMP (shown in the figure) and *c*_0_ = 18 mgC/L (**a**) and at different *c*_0_ (shown in the Figure) and constant TMP = 40 mbar (**b**). Other parameters are presented in Table 1.

**Figure 4 membranes-11-00431-f004:**
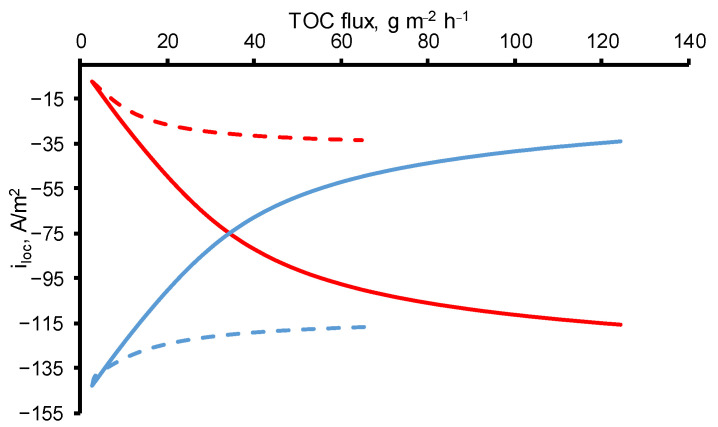
The current densities directed towards the oxygen evolution reaction (blue lines) and *DET* of *OA* reaction (red lines) at a constant initial concentration (*c*_0_ = 18 mgC/L) and different TMP (dashed lines) and at constant TMP (40 mbar) and different initial concentrations (solid lines). Other parameters are presented in Table 1.

**Figure 5 membranes-11-00431-f005:**
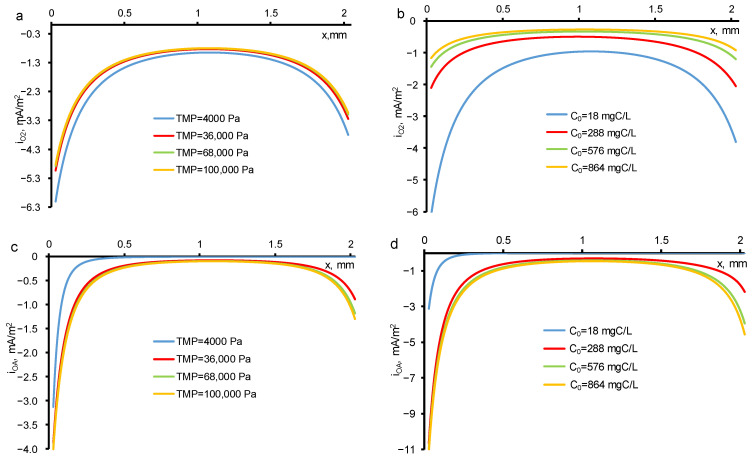
The distribution of local current density directed towards the oxygen evolution reaction (**a**,**b**) and *DET* reaction of *OA* (**c**,**d**) at a constant initial concentration (*c*_0_ = 18 mgC/L) and different TMP (**a**,**c**) and at constant TMP (40 mbar) and different initial concentrations (**b**,**d**). The TMP and *c*_0_ are shown in the figures. Other parameters are presented in Table 1.

**Figure 6 membranes-11-00431-f006:**
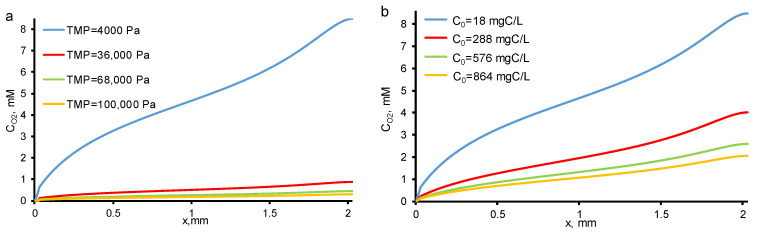
Calculated concentration distributions of oxygen at different TMP (shown in the Figure) and *c*_0_ = 18 mgC/L (**a**) and at different *c*_0_ (shown in the Figure) and constant TMP = 40 mbar (**b**). Other parameters are presented in Table 1.

**Figure 7 membranes-11-00431-f007:**
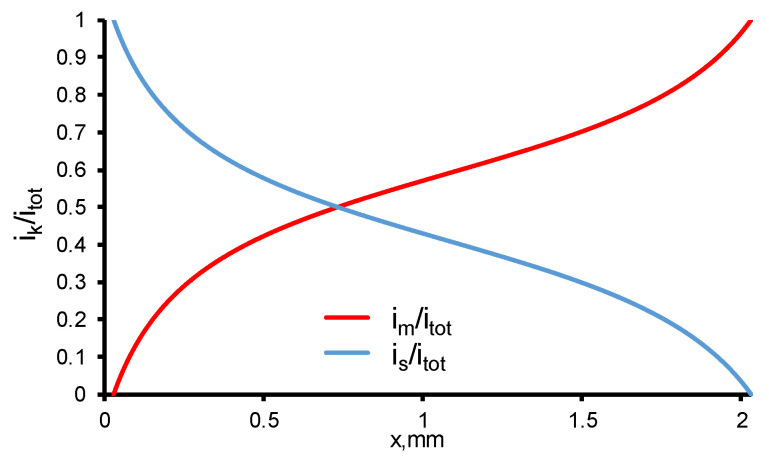
Calculated current densities in REM normalized by *i_tot_* in solution (*i_s_*) and electrode (*i_m_*) material phases at TMP = 40 mbar and *c*_0_ = 18 mgC/L. Other parameters are presented in Table 1.

**Figure 8 membranes-11-00431-f008:**
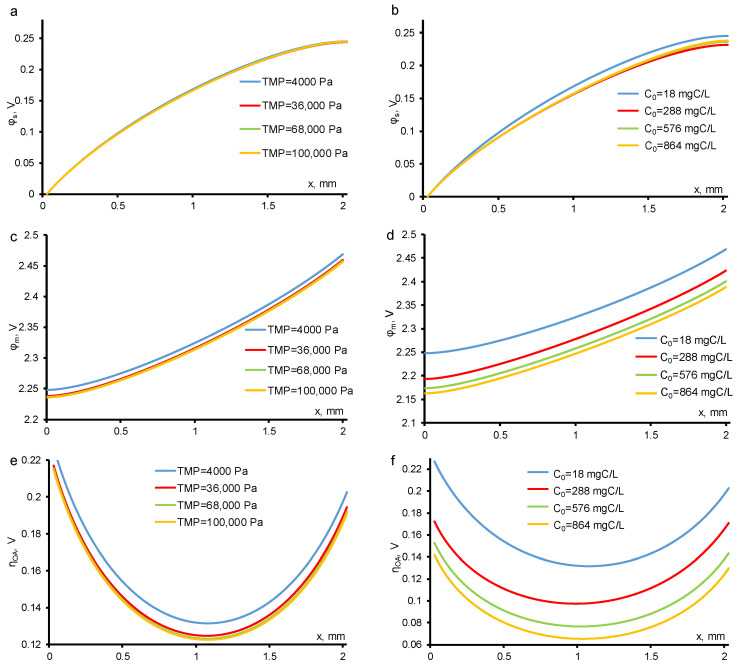
Calculated potential in electrolyte (**a**,**b**) electrode material (**c**,**d**) phases and overpotential for *OA*
*DET* reaction (**e**,**f**) distributions at different TMP (shown in the figure) and *c*_0_ = 18 mgC/L (**a**,**c**,**e**) and at different *c*_0_ (shown in the figure) and constant TMP = 40 mbar (**b**,**d**,**f**). Other parameters are presented in Table 1.

**Figure 9 membranes-11-00431-f009:**
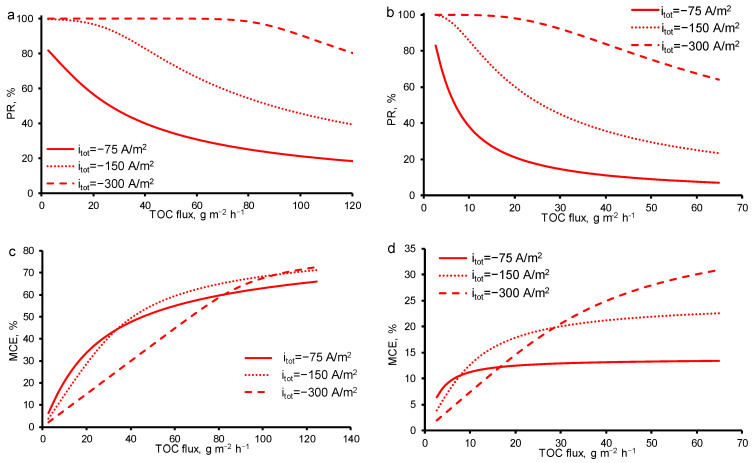
Calculated efficiency of oxalic acid removal from water by anodic oxidation on the REM as a function of *TOC flux* through the porous electrode at different current densities: (**a**,**b**) percentage of *TOC* removal (*PR_TOC_*), (**c**,**d**) mineralization current efficiency (*MCE*). Results from increasing concentrations of oxalic acid (TMP = constant = 40 mbar) (**a**,**c**) are compared with results from increasing TMP (*c*_0_ = constant = 18 mgC/L) (**b**,**d**). The current density is shown in the figures. Other parameters are presented in Table 1.

**Table 1 membranes-11-00431-t001:** Parameters of the system used in the simulations.

Parameter	Definition	Value	Reference
*ε_s_*	fraction volume of solution in REM (porosity)	0.41	[5]
*ε* _m_	fraction volume of electrode material in REM	1 − *ε_s_*	[5]
*D_OA_*	diffusion coefficient of *OA*	1.0 × 10^−9^ m^2^/s	[41]
*D* _O_2__	diffusion coefficient of O_2_	2.0 × 10^−9^ m^2^/s	[42]
$D_{}^{a}$	dispersion coefficient	v × 3 × 10^−4^ m	*
*a* _v_	specific surface area of the electrode	10^8^ 1/m	[5]
$\kappa_{s}$	electrolyte conductivity	1.3 S/m	[5]
$\kappa_{m}$	electrode conductivity	1.3 S/m	
*σ*	permeability coefficient	1.7 × 10^−14^ m^2^	[5]
*μ*	dynamic viscosity	8.9 × 10^−4^ Pa×s	
*d*	REM thickness	2 mm	[5]
$i_{0}^{OA}$	exchange current density of *OA*	−10^−6^ A/m^2^	*
$i_{0}^{O_{2}}$	exchange current density of O_2_	−10^−6^ A/m^2^	*
$c_{0}^{OA}$	concentration of *OA* to which the exchange current density is referred	0.75 mol/m^3^	*
$\beta_{OA}$	electron transferred coefficient in reaction (11)	0.5	*
$\beta_{O_{2}}$	electron transferred coefficient in reaction (10)	0.125	*
$n_{OA}$	number of electrons transferred in reaction (11)	2	
$n_{O_{2}}$	number of electrons transferred in reaction (10)	4	
$E_{OA}^{0}$	formal potential for oxidation of *OA*	2.02 V	*
$E_{O_{2}}^{0}$	formal potential in reaction (11)	1.8 V	*
$s_{OA}$	stoichiometric coefficient of *OA* in reaction (11)	−1	
$s_{O_{2}}$	stoichiometric coefficient of O_2_ in reaction (10)	1	
*i_tot_*	total current density	−150 A/m^2^	[5]
T	temperature	298.15 K	[5]
*δ*	diffusion layer thickness	30 μm	[30]
*k* _m_	mass transfer coefficient	$\;0.91D_{OA}a_{v}{\left(\frac{v}{a_{v}\nu \psi }\right)}^{0.49}\psi^{2}{\left(\frac{v}{D_{OA}}\right)}^{\frac{1}{3}}$	[35]
*ψ*	shape factor	0.86	
*ν*	kinematic viscosity	8.9 × 10^−7^ m^2^/s	

*—fitting parameters.

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
