# Peer review of "A Simple 1D Convection-Diffusion Model of Oxalic Acid Oxidation Using Reactive Electrochemical Membrane"

_membranes, 2021, doi:10.3390/membranes11060431_

Round 1

Reviewer 1 Report

This work concerning the model of a 1D convection-diffusion-reaction in a reactive electrochemical membrane, merit to be published because it favours the advancement of the knowledge in this field. As stated the authors the mathematical modelling in this area is very poorly represented, and this paper contains a well described model based on experimental data related to oxalic acid oxidation.

Observations:

-in Figure 2 caption is opportune clarify that blue graph corresponds to OA and the red one to O2, this can be seen in the Figure 1, but is more adequate report in the Figure 2 caption this specification. Moreover, referring to oxygen graphs in Figure 2 a and b, describe briefly in the main text paragraph the reason why we can observe a marked deviation from theoretical respect experimental plot, even if authors previously indicate that same simplification on the model were done, please indicate in the paragraph the possible motivation of the differences of the two trends especially at higher total organic carbon flux.

-Figure 5 is divided on two page, for a better graphical presentation distribute the text in such way to maintain Figure and caption on the same page, do the same for Figure 7.

-Each image that compose Figure 8 is partially cut and not clearly showed, please try to correct it.

After the indicated minor revisions, I agree for the publication of the paper.

Reviewer 2 Report

This work theoretically analyzed the performance of an electrochemically reactive membrane reactor using a one-dimensional model. This modeling work seems to be well organized, and the manuscript is also well-written. Moreover, the modeling results are in good agreement with experimental data extracted from previous studies; the publication of this research work in this journal is recommended after addressing some questions and revisions.

1) The differentiation and originality of this study against previous studies are not clearly mentioned. A clear explanation on this should be added in the Introduction section.

2) For modeling, there are several assumptions. There are no reasons why these assumptions were adopted mentioned. Reasonable reasons for this should be given in the manuscript.

3) in authors’ previous work, 2D modeling was used to predict the performance of the same system. what is the difference between 1D and 2D modeling? Why did authors conduct theoretical analysis using 1D convection-diffusion model? the comparison of two approaches (1D and 2D model) should be added in the manuscript.

4) Why does oxygen evolution reaction play an important role?

5) Line 147, the position of the reference [36] is not proper.

6) “But in our case, OA has a low oxidation rate by HO• and faster mineralization can be 149

achieved by direct electron transfer (DET). The generated HO• radicals are consumed in 150

the recombination reaction and form the oxygen molecule. “

How does authors make sure this?

7) Text size of Figures 6-9 is too small. To make it more readable, increase the text size.
